# Predictive Power of MIB-1 vs. Mitotic Count on Progression-Free Survival in Skull-Base Meningioma

**DOI:** 10.3390/cancers14194597

**Published:** 2022-09-22

**Authors:** Tim Lampmann, Johannes Wach, Marie-Therese Schmitz, Ági Güresir, Hartmut Vatter, Erdem Güresir

**Affiliations:** 1Department of Neurosurgery, University Hospital Bonn, 53127 Bonn, Germany; 2Department of Medical Biometry, Informatics and Epidemiology, University Hospital Bonn, 53127 Bonn, Germany

**Keywords:** meningioma, skull-base, MIB-1, mitotic count, recurrence, progression-free survival

## Abstract

**Simple Summary:**

Meningiomas are mainly benign intracranial tumors. Nevertheless, risk of recurrence exists in long-term follow-up, so new prognostic markers are still need to be identified. MIB-1 is no diagnostic criterion in WHO classification of meningiomas by now. This retrospective study shows that MIB-1 as well as mitotic count are good predictors for progression-free survival in skull-base meningiomas. The implantation of MIB-1 may enable an improved classification of meningiomas regarding progression-free survival. Moreover, this analysis of skull-base meningiomas shows that current cut-offs may have to be adjusted for meningioma location.

**Abstract:**

Although meningiomas are mainly non-aggressive and slow-growing tumors, there is a remarkable recurrence rate in a long-term follow-up. Proliferative activity and progression-free survival (PFS) differs significantly among the anatomic location of meningiomas. The aim of the present study was to investigate the predictive power of MIB-1 labeling index and mitotic count (MC) regarding the probability of PFS in the subgroup of skull-base meningiomas. A total of 145 patients were included in this retrospective study. Histopathological examinations and follow-up data were collected. Ideal cut-off values for MIB-1 and MC were ≥4.75 and ≥6.5, respectively. MIB-1 as well as MC were good predictors for PFS in skull-base meningiomas. Time-dependent analysis of MIB-1 and MC in prediction of recurrence of skull-base meningioma showed that their prognostic values were comparable, but different cut-offs for MC should be considered regarding the meningioma’s location. As the achievement of a gross total resection can be more challenging in skull-base meningiomas and second surgery implies a higher risk profile, the recurrence risk could be stratified according to these findings and guide decision-making for follow-ups vs. adjuvant therapies.

## 1. Introduction

Although meningiomas—especially World Health Organization (WHO) grade 1 and 2—are mainly non-aggressive and slow-growing tumors, there is a remarkable recurrence rate in a long-term follow-up [1,2]. First-line treatment is complete resection to prevent tumor recurrence [3]. There is growing evidence for increased cellular proliferative potential as an important mechanism of oncogenesis [4]. Recent studies emphasized the importance of Molecular Immunology Borstel 1 (MIB-1) labeling index and mitotic count (MC) as independent predictors of progression-free survival (PFS) [5,6,7,8,9]. However, proliferative activity differs significantly among the anatomic location (non-skull base, skull base, spinal) of meningiomas. Hence, tumor biology and cytoreductive treatment significantly influences the probability of PFS [10]. The aim of the present study was to investigate the predictive power of MIB-1 labeling index and MC regarding the probability of PFS in the subgroup of skull-base meningiomas

## 2. Materials and Methods

### 2.1. Study Design and Patient Characteristics

A total of 880 patients were surgically treated for WHO grade 1 and 2 meningioma between 01/2009 and 07/2019 at our neurosurgical department. Patient data were retrospectively reviewed. Institutional review board approval had been obtained. Histopathologically confirmed meningioma, intracranial localization, age equal or greater than 18 years, the availability of the MIB-1 index and/or MC, and neurosurgical resection were inclusion criteria for this study. Meningiomas outside of skull-base or associated with neurofibromatosis type 2 were excluded due to differences regarding histopathology and proliferation potential [10,11]. Patients with macroscopic residual tumor as patients who underwent a Simpson grade > III resection (constituting a subtotal or partial resection/biopsy) were excluded because those resected tumors do not necessarily contain the “hotspot region”, which reflects the area of maximum proliferative activity [12]. Moreover, patients lost to follow-up were also excluded. In all, 145 patients were included for data analysis (Figure 1).

### 2.2. Data Recording

Clinical information including age, sex, comorbidities, Karnofsky performance status (KPS), body mass index (BMI), peritumoral edema, tumor growth pattern, WHO grading based on postoperative histopathological examination, immunohistochemical examinations, extent of tumor resection based on the Simpson grading system according to the European Association of Neuro-Oncology (EANO), surgical and medical complications according to the classification of Landriel Ibañez [13], and postoperative follow-up data were collected and entered into a computerized database (SPSS, Version 27 for Windows, IBM Corp., Armonk, NY, USA) [14,15]. Meningioma location was subdivided into three groups: medial skull-base, lateral skull-base, and occipital fossa [10].

### 2.3. Histopathology

The following histopathological investigations were previously described [15,16,17]: Histopathological grading was performed based on the 2016 WHO criteria [1]. All pathology reports underwent renewed review to confirm that diagnosis was in keeping with these requirements. Immunohistochemistry was performed in a similar fashion as described before for paraffin-embedded biopsy tissue specimens [18,19]. MIB-1 labeling index was determined using an anti-Ki67 antibody (Clone Ki-67P, dilution 1:1000, DAKO, Glostrup, Denmark). Diaminobenzidine was used for visualization. An expert neuropathologist carried out the investigations. The MIB-1 index was assessed in randomly selected high-power microscopic fields as proportion of stained and unstained nuclei in the neoplastic cells. MC were regularly investigated as “number of mitotic figures/10 high power fields” as a given diagnostic criteria of atypical meningioma [1].

### 2.4. Follow-Up

As institutional routine, clinical and imaging follow-up consisted of MRI scans at 3 months after surgery as well as on an annual basis for the following years [15]. Earlier or shorter intervals of clinical and imaging follow-up were strongly recommended in case of new or worsened neurological deficits as well as radiological signs of meningioma progression or recurrence [15]. Recurrence of tumors were considered for analysis only if tumors occurred at the same location as the initial surgery. The time to recurrence was defined as the time between surgery and radiological recurrence (i.e., date of MRI). Radiological recurrence of tumors without clinical or functional implications, thus not requiring any subsequent therapy, were not included in the analysis [20].

### 2.5. Statistical Analysis

Data were organized and analyzed using SPSS (Version 27, IBM Corp, Armonk, NY, USA). Descriptive statistics of baseline characteristics included mean and standard deviation (SD) for continuous variables and absolute and relative frequencies for categorical variables. Receiver-operating characteristic (ROC) analysis was performed to evaluate how MIB-1 index and MC discriminate between meningioma recurrence. The selection of cut-off points was based on the Youden index [16]. Kaplan–Meier estimates were used to visualize the probability of PFS in either MIB-1 and MC groups. To evaluate the time-varying performance of MIB-1 and MC on PFS time-dependent ROC analysis was conducted using the R package “risksetROC” of the R software (version 4.0.4; R Foundation for Statistical Computing, Vienna, Austria) [21,22].

## 3. Results

### 3.1. Patient Characteristics

In all, 145 patients were included in the present study. The mean age was 59.9 years and the female/male ratio was 2.6:1. The mean KPS before surgery was 90.9. Tumor grading according to the WHO classification criteria [1] classified 126 (86.9%) tumors as grade 1 and 19 (13.1%) tumors as grade 2. Meningiomas were predominantly located at lateral skull-base (49%), followed by medial skull-base (30.3%) and occipital fossa (20.7%). Substantial peritumoral edema was caused by 73 (50.3%) tumors, and 8 (5.5%) patients suffered multiple meningiomas. Simpson grade I/II resections were achieved in 135 (93.1%) patients, whereas 10 (6.9%) patients underwent Simpson grade III resections. MIB-1 index was available in all 145 patients and MC in 117 patients. Those 28 meningiomas in whom MC were not available showed no tumor recurrence. These results are summarized in Table 1.

### 3.2. MIB-1 Labeling Index and Mitotic Count in Prediction of Recurrence of Skull-Base Meningioma

Mean MIB-1 labeling index was 4.9 ± 2.3% and 1.3 ± 2.2 for MC. The AUC of MIB-1 labeling index and MC for intracranial recurrent meningioma were 0.71 (95% CI: 0.58–0.84, *p* = 0.06) and 0.58 (95% CI: 0.38–0.75, *p* = 0.44), respectively. Sensitivity and specificity of MIB-1 labeling index at the threshold of 4.75 (according to Youden index) were estimated to 89% and 46%, respectively. For MC, sensitivity and specificity at the threshold of 6.5 were 22% and 98%, respectively. Figure 2 shows the ROC curves of MIB-1 labeling index and MC, respectively.

Patients with an MIB-1 labeling index of <4.75% had a median time to progression of 456 weeks (95% CI: 431–480) and patients with an MIB-1 labeling index of ≥4.75% had a median PFS time of 402 weeks (95% CI: 358–445). Patients with an MC of <6.5% had a median PFS time of 406 weeks (95% CI: 376–436) and patients with an MC of ≥6.5% had a median PFS time of 250 weeks (95% CI: 71–429) (Figure 3).

### 3.3. Time-Dependent Analysis of MIB-1 Labeling Index and Mitotic Count in Prediction of Recurrence of Skull-Base Meningioma

Results of the time-dependent ROC analysis of MIB-1 and MC are shown in Table 2 as well as Figure 4. The C-index that represents the probability that MIB-1 or MC in a patient with a shorter time to tumor progression after surgery is high, was estimated to 0.61 (95% CI 0.51–0.77) for MIB-1 and 0.60 (95% CI 0.50–0.81) for MC, respectively.

## 4. Discussion

Primary treatment of cranial meningioma is a maximum cytoreductive approach, that mostly results to be curative [3]. In certain cases tumor recurrence is observed if the follow-up is long enough [2]. Current follow-up involves yearly MRI scans and sometimes even shorter intervals, if tumor was incompletely removed or recurred [3]. Therefore, it is important to identify predictors for meningioma recurrence, so that either follow-up could be optimized or adjuvant therapy could be advocated. WHO grading aims to classify the risk of recurrence. In the current as well as previous version of WHO classification MC and brain invasion are major diagnostic criteria of atypical meningioma and therefore predictors [1,23].

Nevertheless, there is still an incongruence between the WHO grade and clinical course, so new prognostic markers are still need to be identified [24]. MIB-1 and MC are already known predictors for tumor recurrence, but only MC is integrated in the current classification as described above [5,6,7,8,9,14,23].

MIB-1 stains nuclear protein Ki67 which is expressed in proliferation phases of the cell cycles [25]. It’s bearing as an additional criterion for meningioma grading is controversial [24]. By now, use of MIB-1 is more common in breast cancer, but there are differences in procedure leading to inter- and intralaboratory variability [26]. Some laboratories or investigators use hot spot analysis, where number of positive nuclei were divided by the total number of nuclei within the hot spot, other laboratories use global scoring, where entire tumor area was analyzed [26]. In addition, cut-offs are not standardized, so laboratory-specific cut-offs are maybe necessary [27]. This inter- and intralaboratory variability causes that no generalized recommendation can be made for the use of MIB-1 in meningiomas. MC were defined as “number of mitotic figures/10 high power fields” as stated above [1]. These mitotic figures were usually counted by a neuropathologist on hematoxylin and eosin-stained slides just like in our cohort. This requires a neuropathologist’s expertise and carries therefore poor intra- and interobserver reproducibility [28]. An additional staining with phosphohistone H3 could improve reproducibility, but the threshold of MC should be adopted when utilizing new methods [28]. Therefore, both MIB-1 and MC are limited in validity due to their different determinations. One possibility to overcome this issue would be a multicentric trial investigating the predictive power of the MIB-1 labeling index regarding PFS. Moreover, a future implementation in the classification system necessitates a homogenous approach regarding the determination of MIB-1 labeling index and has to be investigated in such a multicentric trial (e.g., hotspot method, digital image analysis, average method).

The static allocation of all meningiomas regardless of their origins—e.g., skull-base, non-skull-base and spinal—appears to be inaccurate as the location has considerable impact on meningioma PFS, especially in skull-base meningiomas [29,30]. In the present study, we compared MIB-1 and MC for their predictive power in skull-base meningiomas.

Intuitively, MC was associated with meningioma recurrence but with a slightly higher cut-off as in the WHO classification [1]. MIB-1 was associated with meningioma recurrence with a cut-off similar to previous studies [16,17,24,31]. So both MC and MIB-1 are valuable predictors for PFS in our cohort. However, other studies demonstrated MIB-1 to be a marker for time to recurrence rather than a predictor of recurrence [24]. We therefore analyzed time-dependent ROC curves to reveal time differences. AUCs reached their maximum on week 183 with 0.63 for MC and 0.64 for MIB-1, respectively. Time-dependent AUCs did not differ significantly inter- and intraindividually, while MC of ≥6.5% had a median PFS time of 250 weeks in contrast to 402 weeks for MIB-1 of ≥4.75%.

Although MIB-1 is not a classification criterion by now, many neuropathologists determine MIB-1 already as a routine. This study should not induce to replace MC by MIB-1 but to use it in addition. We suggest that in the modern era of tailored neuro-oncological care a maximum of potential molecular markers may be used to achieve a most efficient and reliable individual treatment as well as follow-up scheduling. The additional assessment of MIB-1 regarding tumor recurrence risk could improve stratification of patients to specific follow-up. Nevertheless, this study also emphasizes that different cut-offs for both, MIB-1 and MC, may be necessary in order to comply with the different pathophysiology depending on meningioma location.

## 5. Conclusions

MIB-1 as well as MC are good predictors for PFS in skull-base meningiomas. Their prognostic value is more or less equal. We do not want to substitute MC with MIB-1, rather emphasize the possible benefit of MIB-1 as an additional histopathological factor. Moreover, higher cut-off for MC should be applied in predicting PFS among skull-base meningiomas, but further studies are needed.

As the achievement of a gross total resection can be more challenging in skull-base meningiomas and second surgery implies a higher risk profile, the recurrence risk could be stratified according to these findings and guide decision-making for follow-ups vs. adjuvant therapies.

## Figures and Tables

**Figure 1 cancers-14-04597-f001:**
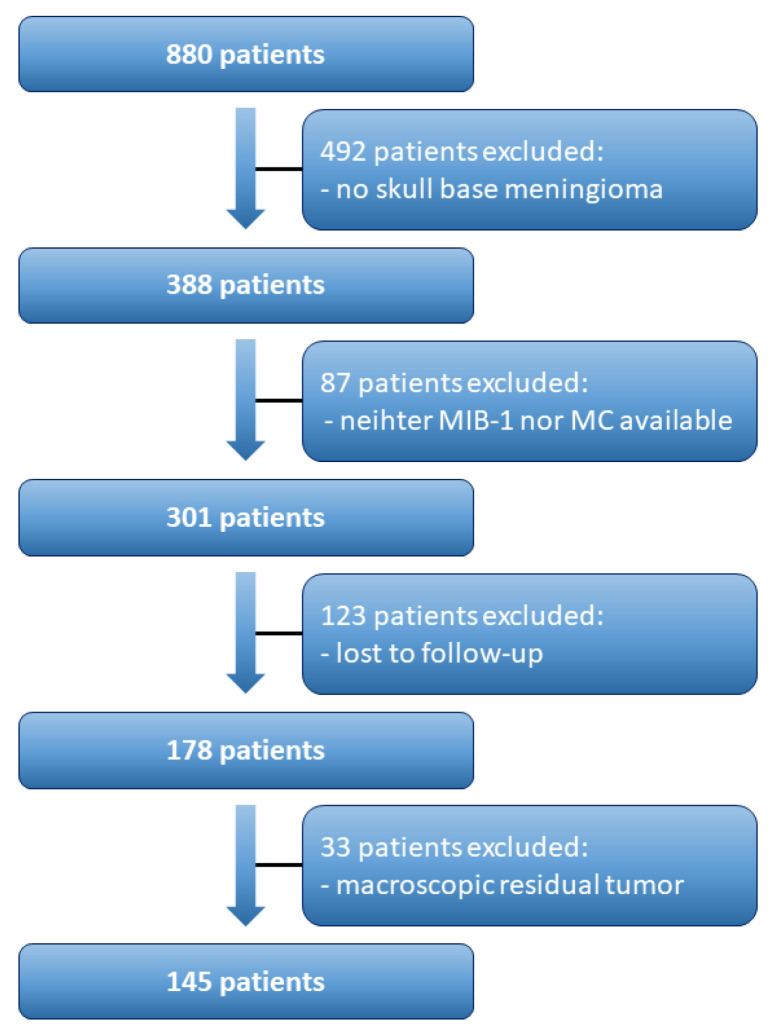
Flow chart illustrating the selection process of consecutive meningioma patients.

**Figure 2 cancers-14-04597-f002:**
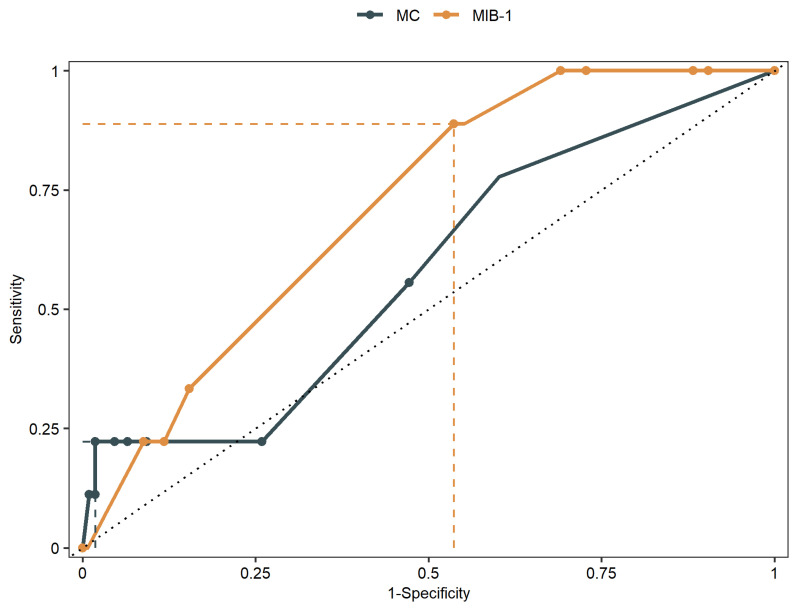
Receiver-operating characteristic curves illustrating MIB-1 labeling index and MC in prediction of PFS of skull-base meningiomas.

**Figure 3 cancers-14-04597-f003:**
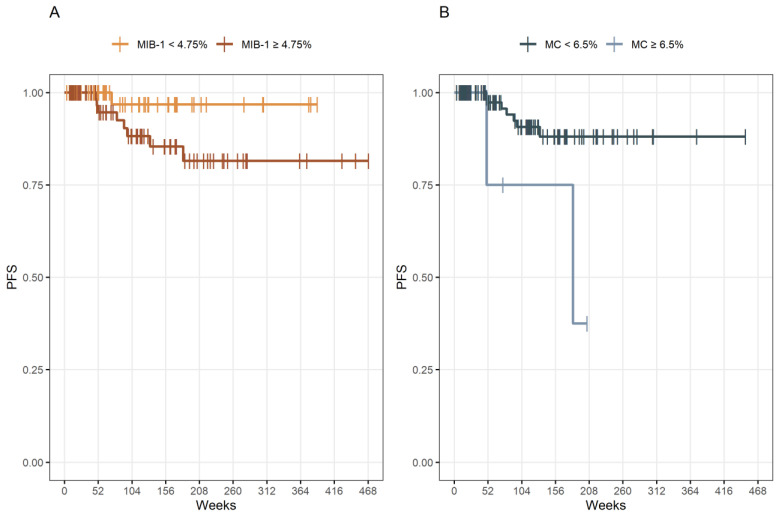
(**A**) Kaplan–Meier analysis of PFS stratified by “MIB-1 ≥ 4.75%” (dark yellow line) and “MIB-1 < 4.75%” (yellow line), *p* = 0.083. (**B**) Kaplan–Meier analysis of PFS stratified by “MC ≥ 6.5” (blue line) and “MC < 6.5” (dark blue line), *p* = 0.014. Vertical dashes indicate censored data (here: progression-free at last follow-up) within the PFS curves.

**Figure 4 cancers-14-04597-f004:**
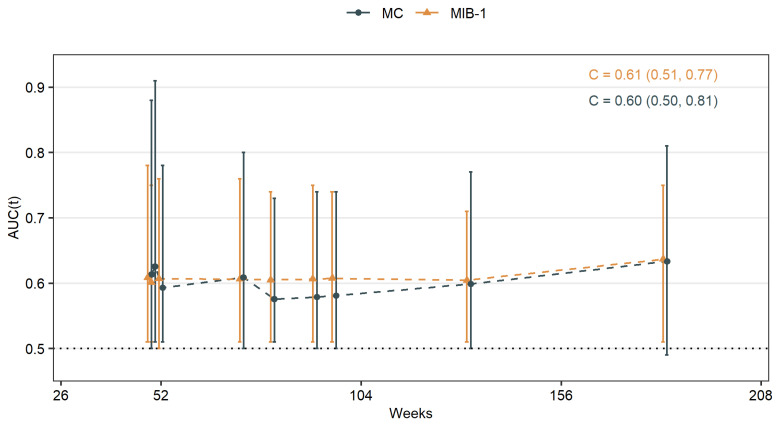
Time-dependent AUCs for MIB-1 labeling index and MC in skull-base meningiomas.

**Table 1 cancers-14-04597-t001:** Baseline characteristics (n = 145).

Mean Age (±SD) [in Years]	59.9 ± 13.2
Sex	
Female	105 (72.4%)
Male	40 (27.6%)
Mean preoperative KPS (±SD)	90.9 ± 11.0
WHO grade	
1	126 (86.9%)
2	19 (13.1%)
Tumor location	
Medial skull-base	44 (30.3%)
Lateral skull-base	71 (49%)
Occipital fossa	30 (20.7)
Multiple meningiomas	8 (5.5%)
Sinus invasion	27 (18.6%)
Peritumoral edema	73 (50.3%)
Simpson grade	
Simpson grade I & II	135 (93.1%)
Simpson grade III	10 (6.9%)
Availability of	
MIB-1	145
MC	117
Mean MIB-1 (±SD) (in %)	4.9 ± 2.3
WHO grade 1	4.8 ± 2.2
WHO grade 2	5.4 ± 2.5
High MC (≥4)	9/117 (7.7%)
WHO grade 1	5/102 (4.9%)
WHO grade 2	4/15 (26.7%)
Mean MC (±SD)	1.3 ± 2.2
WHO grade 1	1.0 ± 1.6
WHO grade 2	3.0 ± 4.4
Mean Follow-Up (±SD) (in weeks)	116 ± 106
recurrence rate	9/145 (6.2%)
WHO grade 1	7/126 (5.6%)
WHO grade 2	2/19 (10.5%)
Landriel Ibañez Classification	
None	114 (78.6%)
Grade I a	2 (1.4%)
Grade I b	10 (6.9%)
Grade II a	1 (0.7%)
Grade II b	9 (6.2%)
Grade III a	5 (3.4%)
Grade III b	4 (2.8%)
Grade IV	0 (0%)

**Table 2 cancers-14-04597-t002:** Time-dependent AUCs for PFS skull-base meningiomas.

PFS (weeks)	AUC	
	MIB-1	MC
49	0.61	0.61
50	0.60	0.63
52	0.61	0.59
73	0.61	0.61
81	0.60	0.58
92	0.61	0.58
97	0.61	0.58
132	0.60	0.60
183	0.64	0.63
**C-index**	0.61	0.60

## Data Availability

The data presented in this study are available in this article.

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
