# Peer review of "Predictive Power of MIB-1 vs. Mitotic Count on Progression-Free Survival in Skull-Base Meningioma"

_cancers, 2022, doi:10.3390/cancers14194597_

Round 1

Reviewer 1 Report

This manuscript demonstrates that MIB-1 labeling index and mitotic count are equally useful in predicting the recurrence of skull base meningioma after resection. The authors especially stress the higher cut-off for mitotic count (= 6.5) as a predictor for recurrence compared with non-skull base meningioma. Although the topic addressed is interesting, much improvement can still be made to make it clearer.

 1.     In the abstract, the authors stated “patient data of 880 patients were retrospectively reviewed”. However, only 145 patients were eventually included in this study. I suspect that the authors should have expressed the real number of patients included here.

2.     Was there any patient who underwent adjuvant radiotherapy after resection in this cohort? According to the authors, “radiological recurrence without clinical or functional implications were not included in the analysis”. We often encounter the enlargement of residual tumors after resection without any clinical manifestation, resulting in preventive radiotherapy before further enlargement. Did this cohort include such cases?

3.     In the Conclusions, the authors stated that “different cut-offs for MC should be considered regarding the meningioma’s location.” This sentence might be a little misleading. Did the authors analyze the PFS by sub-types among skull base meningiomas? (ie. Medial skull base/ lateral skull base /occipital fossa) Maybe they did not, because any result is not shown. Thus, I wonder if it is more appropriate like this: “Higher cut-off for MC should be applied in predicting PFS among skull-base meningiomas.”

4.     As the authors mentioned, the inter- and intralaboratory variability cause that no recommendation can be made for the use of MIB-1 index so far. How could the authors overcome this problem through the study? Namely, how can we utilize the result of this study outside their institution?

Author Response

Thank you for the useful comments. We believe that after incorporating the issues listed below, the manuscript is clearer and the study is now better understood. We hope that the manuscript now is eligible for publication.

  1. In the abstract, the authors stated “patient data of 880 patients ~ were retrospectively reviewed”. However, only 145 patients were eventually included in this study. I suspect that the authors should have expressed the real number of patients included here.

You are right; we revised the abstract to clarify the included patient number: “145 patients were included in this retrospective study.”

  1. Was there any patient who underwent adjuvant radiotherapy after resection in this cohort? According to the authors, “radiological recurrence without clinical or functional implications were not included in the analysis”. We often encounter the enlargement of residual tumors after resection without any clinical manifestation, resulting in preventive radiotherapy before further enlargement. Did this cohort include such cases?

We also observed enlargement of residual tumors in patients at our institution (included in the initial cohort consisting of 880 patients), but we aimed to analyze a homogenous cohort by excluding residual tumors (“Patients with macroscopic residual tumor […] were also excluded”). We therefore revised section “2.1. Study Design and Patient Characteristics” as follows: “Patients with macroscopic residual tumor as patients who underwent a Simpson grade >III resection (constituting a subtotal or partial resection / biopsy) were excluded because those resected tumors do not necessarily contain the “hotspot region”, which reflects the area of maximum proliferative activity [12]. Moreover, patients lost to follow-up were also excluded.” Hence, an inclusion of not completely removed meningiomas would have been a confounder regarding our primary endpoint “probability of progression-free survival” (https://doi.org/10.1097/00005072-199311000-00008). A residual tumor or a secondarily applied preventive radiotherapy would have influenced the analysis of MIB-1 and MC regarding the probability of progression-free survival. However, we absolutely agree with the reviewer regarding the impact of MIB-1 or MC in subtotally resected meningiomas, which underwent a preventive radiotherapy prior to a clinic-radiological correlation. This issue is of paramount importance but the power size of our institutional series is too low to enable a reliable analysis. 

  1. In the Conclusions, the authors stated that “different cut-offs for MC should be considered regarding the meningioma’s location.” This sentence might be a little misleading. Did the authors analyze the PFS by sub-types among skull base meningiomas? (ie. Medial skull base/ lateral skull base /occipital fossa) Maybe they did not, because any result is not shown. Thus, I wonder if it is more appropriate like this: “Higher cut-off for MC should be applied in predicting PFS among skull-base meningiomas.”

A detailed analysis of the different sub-sub-types (ie. medial skull base/ lateral skull base /occipital fossa) was not performed, as the sample size would have been too small to detect significantly differences (e.g., MC ≥4 occurred only in 9 patients of the whole cohort of 117 patients). The reviewer is absolutely right that the stated phrase in the section “5. Conclusions” is not entirely based on our results. Hence, we have revised this sentence according to your suggestion and replaced it by the following: “Moreover, higher cut-off for MC should be applied in predicting PFS among skull-base meningiomas, but further studies are needed.”

  1. As the authors mentioned, the inter- and intralaboratory variability cause that no recommendation can be made for the use of MIB-1 index so far. How could the authors overcome this problem through the study? Namely, how can we utilize the result of this study outside their institution?

The reviewer is absolutely right that heterogenous methods in the determination of the MIB-1 labeling index substantially influence the current evidence of MIB-1 labeling index regarding probability of progression-free survival. Hence, our institution is part of a multicentric approach investigating the predictive power of the MIB-1 labeling index regarding PFS in primary sporadic cranial meningiomas. This multicentric study will consider three different approaches (hotspot method, digital image analysis, average method) of MIB-1 labeling index determination. However, we strive to finalize this study at the end of the year and hope that this multicentric investigation will provide more insight into this important issue regarding inter- and intralaboratory variability in MIB-1 labeling index determination of meningiomas.

We revised the section “4. Discussion” to clarify these possible future implementations and emphasize issues which still have to be analyzed: “One possibility to overcome this issue would be a multicentric trial investigating the predictive power of the MIB-1 labeling index regarding PFS. Moreover, a future implementation in the classification system necessitates an homogenous approach regarding the determination of MIB-1 labeling index and has to be investigated in such a multicentric trial (eg., hotspot method, digital image analysis, average method).”

Reviewer 2 Report

The authors should focus more on the clinical relevance of the statistical correlations.

The surgical aspects has to be considered.

Different groups of lesions according to the location has be presented.

Author Response

Thank you for the useful comments. We believe that after incorporating the issues listed below, the manuscript is clearer and the study is now better understood. We hope that the manuscript now is eligible for publication.

The authors should focus more on the clinical relevance of the statistical correlations.

We suggest that a combined approach using both MIB-1 labeling index and mitotic count might enable a more tailored approach regarding the risk stratification of progression-free survival. For instance, in a retrospective study of 239 WHO grade 1 (constituting MC <4) meningiomas it was found that the recurrence rate was 18.8% in those patients who underwent a gross total resection of a meningioma with an MIB-1 labeling index >4.5%. Conversely, the WHO grade 1 meningioma patients who underwent a subtotal resection of a meningioma with an MIB-1 labeling index <4% had the same risk of recurrence (https://doi.org/10.3389/fonc.2020.01522). Hence, the MIB-1 labeling index might have an additional predictive value to the established determination of the MC as far as risk stratification of tumor progression is concerned.

The surgical aspects has to be considered.

 We reviewed all 145 patients and gathered all surgical and medical complications according to the classification of Landriel Ibañez during treatment course. We revised section “2.2. Data Recording”: “Clinical information including age, sex, comorbidities, Karnofsky performance status (KPS), body mass index (BMI), peritumoral edema, tumor growth pattern, WHO grading based on postoperative histopathological examination, immunohistochemical examinations, extent of tumor resection based on the Simpson grading system according to the European Association of Neuro-Oncology (EANO), surgical and medical complications according to the classification of Landriel Ibañez [13], and postoperative follow-up data were collected and entered into a computerized database (SPSS, Version 27 for Windows, IBM Corp., Armonk, NY, USA) [14,15].”

We amended “Table 1. Baseline characteristics (n=145)” by the results of the new investigation of surgical and medical complications:

Mean Age (±SD) [in years]

59.9 ± 13.2

Sex

Female

Male

105 (72.4%)

40 (27.6%)

Mean preoperative KPS (±SD)

90.9 ± 11.0

WHO grade

1

2

126 (86.9%)

19 (13.1%)

Tumor location

Medial skull-base

Lateral skull-base

Occipital fossa

44 (30.3%)

71 (49 %)

30 (20.7)

Multiple meningiomas

8 (5.5%)

Sinus invasion

27 (18.6%)

Peritumoral edema

73 (50.3%)

Simpson grade

Simpson grade I & II

Simpson grade III

135 (93.1%)

10 (6.9%)

Availability of

MIB-1

MC

145

117

Mean MIB-1 (±SD) (in %)

WHO grade 1

WHO grade 2

4.9 ± 2.3

4.8 ± 2.2

5.4 ± 2.5

High MC (≥4)

WHO grade 1

WHO grade 2

9/117 (7.7%)

5/102 (4.9%)

4/15 (26.7%)

Mean MC (±SD)

WHO grade 1

WHO grade 2

1.3 ± 2.2

1.0 ± 1.6

3.0 ± 4.4

Mean Follow-Up (±SD) (in weeks)

116 ± 106

recurrence rate

WHO grade 1

WHO grade 2

9/145 (6.2%)

7/126 (5.6%)

2/19 (10.5%)

Landriel Ibañez Classification

None

114 (78.6%)

Grade Ia

2 (1.4%)

Grade Ib

10 (6.9%)

Grade IIa

1 (0.7%)

Grade IIb

9 (6.2%)

Grade IIIa

5 (3.4%)

Grade IIIb

4 (2.8%)

Grade IV

0 (0%)

Different groups of lesions according to the location has be presented.

A detailed analysis of the different sub-sub-types (ie. medial skull base/ lateral skull base /occipital fossa) was not performed, as the sample size would have been too small to detect significantly differences (e.g., MC ≥4 occurred only in 9 patients of the whole cohort of 117 patients). Therefore, we have revised the section “5. Conclusions” to be more specific why we did not analyze those sub-sub-types: “Moreover, higher cut-off for MC should be applied in predicting PFS among skull-base meningiomas, but further studies are needed.”

Round 2

Reviewer 2 Report

There have been significant improvements.

Yet, clinical and surgical considerations are poor. Which would this paper add ? Which are the aspects that could be useful in management of meningioma?

Whether the sample is too small having a unique definition of skull base weaken the scientific strength and significance.

Author Response

Thank you again for the useful comments.

You are right that this paper adds low direct clinical implication by now. Aim of this study was to emphasize some points: 1) MIB-1 as well as MC are good predictors for PFS in skull-base meningiomas. As MIB-1 is no diagnostic criterion in WHO classification by now, it may be implanted in future and might enable an improved classification of meningiomas regarding the PFS. 2) But this sub-group analysis of skull-base meningiomas also shows that the current cut-offs (mitotic index: 4/10 high power field) for all meningioma locations (i.e., skull-base and not skull-base) may have to be adjusted. Hence, there might be the need to provide tailored thresholds of mitotic count regarding the anatomic location of each meningioma 3) To determine such new cut-offs, a standardized approach of MIB-1 and MC investigation (cf., inter- and intralaboratory variability) has to be established. We also realized that the sample size of 145 patients may be too small for more sub-sub-group analyses but was powerful for the whole sub-group of skull-base meningiomas. To overcome this issue of small sample size our institution is part of a multicentric approach investigating the predictive power of the MIB-1 labeling index regarding PFS in primary sporadic cranial meningiomas. This multicentric study will consider three different approaches (hotspot method, digital image analysis, average method) of MIB-1 labeling index determination. However, we strive to finalize this study at the end of the year and hope that this multicentric investigation will provide more insight into this important issue regarding inter- and intralaboratory variability in MIB-1 labeling index determination of meningiomas. To date, MIB-1 labeling index can be at least used as an add-on to the present WHO classification and MIB-1 labeling index might enable a more tailored scheduling of follow-up MR-imaging to identify recurrent meningiomas as soon as possible. An enlarged sample size will make it possible to conduct powerful sub-sub-group analyses and add more information as well as guidelines for future meningioma diagnosis and treatment.

We revised section “4. Discussion” (line 184 ff, line 209 ff) and “5. Conclusions” (line 217 f) to add more information about further implications.